# Desmopressin treatment combined with clotting factor VIII concentrates in patients with non-severe haemophilia A: protocol for a multicentre single-armed trial, the DAVID study

Lisette M Schütte,[1] Marjon H Cnossen,[2] Reinier M van Hest,[3] Mariette H E Driessens,[4] Karin Fijnvandraat,[5,6] Suzanne Polinder,[7] Erik A M Beckers,[8] Michiel Coppens,[9] Jeroen Eikenboom,[10] Britta A P Laros-van Gorkom,[11] Karina Meijer,[12] Laurens Nieuwenhuizen,[13] Evelien P Mauser-Bunschoten,[14] Frank W G Leebeek,[1] Ron A A Mathôt,[3] Marieke J H A Kruip[1]

For numbered affiliations see end of article.

**Correspondence to**
Dr Marieke J H A Kruip;
m.kruip@erasmusmc.nl

## ABSTRACT

**Introduction** Haemophilia A is an inherited bleeding disorder characterised by factor VIII (FVIII) deficiency. In patients with non-severe haemophilia A, surgery and bleeding are the main indications for treatment with FVIII concentrate. A recent study reported that standard dosing frequently results in FVIII levels (FVIII:C) below or above FVIII target ranges, leading to respectively a bleeding risk or excessive costs. In addition, FVIII concentrate treatment carries a risk of development of neutralising antibodies. An alternative is desmopressin, which releases endogenous FVIII and von Willebrand factor. In most patients with non-severe haemophilia A, desmopressin alone is not enough to achieve FVIII target levels during surgery or bleeding. We hypothesise that combined pharmacokinetic (PK)-guided administration of desmopressin and FVIII concentrate may improve dosing accuracy and reduces FVIII concentrate consumption.

**Methods and analysis** In the DAVID study, 50 patients with non-severe haemophilia A (FVIII:C ≥0.01 IU/mL) with a bleeding episode or undergoing surgery will receive desmopressin and FVIII concentrate combination treatment. The necessary dose of FVIII concentrate to reach FVIII target levels after desmopressin administration will be calculated with a population PK model. The primary endpoint is the proportion of patients reaching FVIII target levels during the first 72 hours after start of the combination treatment. This approach was successfully tested in one pilot patient who received perioperative combination treatment.

**Ethics and dissemination** The DAVID study was approved by the medical ethics committee of the Erasmus MC. Results of the study will be communicated trough publication in international scientific journals and presentation at (inter)national conferences.

**Trial registration number** NTR5383; Pre-results.

## Strengths and limitations of this study

- ► The DAVID-study is a multicentre, prospective trial including patients from eight haemophilia treatment centres in the Netherlands.
- ► Desmopressin and factor VIII (FVIII) concentrate combination treatment is an innovative treatment option for patients with non-severe haemophilia A.
- ► By using maximum a posteriori Bayesian estimation based on an integrated population pharmacokinetic model and measured FVIII levels, the dosing of FVIII concentrate may be improved, with less FVIII levels above and below target.
- ► The DAVID study is a single-armed trial.
- ► The study population is heterogeneous as all types of surgery and bleeding episodes may be included.

## INTRODUCTION

Haemophilia A is an X-linked bleeding disorder characterised by a factor VIII (FVIII) deficiency. Patients with non-severe haemophilia A (FVIII:C ≥0.01 IU/mL) suffer from bleeding in the perioperative setting and after (minor) trauma. Treatment consists of either desmopressin or FVIII concentrate.

FVIII concentrate is an effective but expensive treatment option. With current dosing based on body weight, baseline FVIII:C and target FVIII:C, FVIII:C both below (7%–45%) and above (32%–81%) FVIII target levels have been observed.[1 2] FVIII:C below target may lead to an increased bleeding risk, whereas levels above target increase the costs. Excessively high FVIII:C may also be associated with thrombosis.[3–5] In addition, high FVIII concentrate doses may induce the development of FVIII neutralising antibodies, causing ineffectiveness of treatment with FVIII concentrate. In cases where antibodies cross-react with

patient's endogenous FVIII, they can even cause a severe phenotype with spontaneous bleeding.[6–8] Therefore, dosing within the target range with restriction of FVIII concentrate use is important.

An alternative to FVIII concentrate is desmopressin, a synthetic analogue of vasopressin. It releases von Willebrand factor (VWF) from the endothelium and FVIII, thereby improving haemostasis.[9–11] Although treatment is often effective, FVIII:C response to desmopressin exhibits a high variability between patients with haemophilia A. Contributing factors reported in the literature are age, baseline FVIII:C and the *F8*-gene mutation.[12–14] However, these factors do not explain the observed variability entirely. In addition, in most patients, FVIII:C does not increase sufficiently to prevent perioperative bleeding.

Other limitations of desmopressin are the experienced side effects and tachyphylaxis. Most side effects are mild and transient, such as vasodilation. More severe side effects like hyponatraemia are rare and can usually be prevented by a fluid restriction.[15] Tachyphylaxis occurs when repeated dosages of desmopressin are given with short time intervals (12–24 hours). The decrease in FVIII:C response is approximately 30% from the second dose onwards in case of a 24-hour interval and is believed to be caused by a temporary depletion of VWF and FVIII from the endothelium.[16]

Combined administration of desmopressin and FVIII concentrate may be able to overcome several of the drawbacks of both separate treatment options. However, there is a lack of experience and knowledge with regard to the efficacy and safety of combination treatment. Moreover, optimisation of dosing is necessary to get and keep FVIII:C within the target range.

A valuable approach could be pharmacokinetic (PK)-guided dosing, where FVIII:C responses of future dosages are predicted based on a population PK model in combination with a limited number (two to three) of FVIII:C measurements, obtained after administration of desmopressin and/or FVIII concentrate. This technique was previously shown to be effective in both the prophylactic and perioperative setting in patients with severe and moderate haemophilia A.[17 18] A prospective trial using PK-guided dosing in the perioperative treatment of patients with moderate and severe haemophilia A is ongoing.[19]

Therefore, we hypothesise that combined PK-guided dosing of desmopressin and FVIII concentrate may be a feasible treatment option in patients with non-severe haemophilia A with a bleeding episode or undergoing surgery. Our aim is to show that the proportion of patients with FVIII:C within the FVIII target ranges can be increased by the use of the combination treatment and PK-guided dosing. In addition, the consumption of FVIII concentrate will be reduced. This innovative approach in haemophilia treatment may be a promising alternative with an increase in quality of care and a concomitant cost reduction.

## METHODS AND ANALYSIS
### Objectives
The primary objective of the DAVID study is to evaluate if combination treatment of desmopressin and PK-guided dosing of FVIII concentrate is able to increase the proportion of patients with non-severe haemophilia A with FVIII:C within the target range during the first 72 hours after start of combination treatment compared with historical controls.

### Secondary objectives
1. To assess FVIII concentrate consumption in all patients.
2. To acquire data to improve the integrated population PK model for desmopressin and FVIII concentrate combination treatment.
3. To establish (possible) adverse events of combination treatment, for example, side effects of desmopressin, bleeding episodes, development of neutralising antibodies and thrombotic events.
4. To evaluate the extent of tachyphylaxis after desmopressin treatment.
5. To perform an economical evaluation to quantify the potential cost reduction of the combination treatment.
6. To evaluate the experienced quality of patient care.

### Design
The DAVID study is a multicentre single-arm trial in patients with non-severe haemophilia A with a bleeding episode or in need of a surgical procedure. The study was designed by a project committee existing of members of all participating sites (see online supplementary data 1 and 2). Patients will receive combination treatment of desmopressin and FVIII concentrate instead of the conventional treatment which consists of FVIII concentrate monotherapy.

### Patient selection and recruitment
Patients with non-severe haemophilia A ≥12 years of age with a bleeding episode or undergoing a surgical procedure, requiring FVIII replacement therapy and having a sufficient response to desmopressin, will be included. Patients will be recruited from one of the following haemophilia treatment centres in the Netherlands: Erasmus University Medical Centre Rotterdam, Academic Medical Centre Amsterdam, Leiden University Medical Centre, University Medical Centre Groningen, Radboud university medical centre Nijmegen, University Medical Centre Utrecht, Maxima Medical Centre Eindhoven and Maastricht University Medical Centre (see also online supplementary data 2). They will be approached by telephone or during a visit to the (outpatient) clinic if they present with a bleeding or need a surgical procedure.

### Inclusion criteria
► Patients with non-severe haemophilia A (FVIII:C≥0.01 IU/mL)
► Requiring a surgical procedure or having a bleeding episode

- ► Requiring replacement therapy with FVIII concentrate for at least 48 hours
- ► Age between 12 and 70 years at study inclusion date
- ► (Parental) informed consent
- ► Results of a desmopressin test available (minimal absolute FVIII:C increase >0.2 IU/mL)

### Exclusion criteria

- ► Patients with other congenital or acquired haemostatic abnormalities
- ► Inadequate response to desmopressin (absolute increase in FVIII:C<0.2 IU/mL) during a previous desmopressin test
- ► FVIII neutralising antibodies (in medical history), unless successfully treated with immunotolerance induction therapy
- ► Initiation of FVIII concentrate treatment >24 hours before study inclusion
- ► Patients not eligible for desmopressin treatment due to contraindications, for example, intolerance, interactions with comedication or due to type of surgery

## Interventions and study procedures

All patients will receive combination treatment of desmopressin and FVIII concentrate during a bleeding episode or in the perioperative setting for at least 48 hours. All patients will receive a standard dose of desmopressin intravenously (0.3 µg/kg; no maximum dose). In order to combine both medication regimens, an individualised dosing advice for FVIII concentrate will be provided by the clinical pharmacologist. The initial dosing advice will be based on the FVIII:C response observed after a test administration of desmopressin (see below) and previously collected patient and population pharmacokinetic data after administration of desmopressin (endogenous FVIII) and FVIII concentrate (exogenous FVIII). The treating physician states the duration of combination treatment and the mode of administration of FVIII concentrate (continuous or intermittent administration) and determines FVIII target ranges. During combination treatment, FVIII:C will be assessed regularly. Accordingly, dose adjustments for FVIII concentrate will be made iteratively based on the results of a desmopressin test administration and using Bayesian analysis. Bayesian analysis will be performed with the NONMEM software using a dedicated integrated population model describing the PK of FVIII following both the administration of desmopressin and FVII concentrate. As bleeding or acute surgery calls for immediate treatment and constructing a dosing advice takes time, a patient may be included in the DAVID study until 24 hours after the start of FVIII concentrate monotherapy.

Adherence to the study protocol will be improved by the use of a separate script per included patient, in which an approximated timeline, contact details and all responsibilities of the involved research and treatment team are written down.

### Concomitant treatment

As patients receive desmopressin, they will have a fluid restriction of 1.5 L per 24 hours, until 24 hours after the last desmopressin administration. Furthermore, all concomitant treatment and therapeutics are allowed, except for the therapeutics specified in the exclusion criteria. This means other treatment than desmopressin or factor concentrate necessary to prevent or treat bleeding, such as tranexamic acid, is allowed as well. The same will apply for prophylactic treatment to prevent thrombosis. These treatment options will be applied at the discretion of the treating physician and will be documented for all patients.

### Desmopressin test administration

All patients must have undergone a desmopressin test with at least three FVIII:C measurements. If performed during childhood (<18 years), the test is only admissible when performed ≤4 years before study inclusion. When no desmopressin test has been performed meeting these criteria, the test should be performed with a standard intravenous desmopressin dose of 0.3 µg/kg infused over 30 min with a minimum of three FVIII:C measurements: before the administration of desmopressin, around 1 hour after desmopressin administration for peak measurement and at least one sample thereafter, for example, after 4 hours. All time points of blood sampling and end of desmopressin administration should be documented precisely, as well as exact desmopressin dose and duration of infusion.

### PK-guided dosing

In order to provide an individualised dosing advice, a Bayesian analysis will be performed on the basis of an integrated population PK model. This population model has been constructed based on two population pharmacokinetic models: (1) a population PK model of FVIII:C response after desmopressin administration[20] and (2) a population PK model of FVIII:C after administration of FVIII concentrate.[20] This latter model was constructed based on perioperative data from 29 adults patients with non-severe haemophilia A from whom 245 FVIII:C measurements were available. The final model estimated the baseline FVIII:C to which estimated FVIII:C, that followed administration of FVIII concentrate, were added according to a one-compartment model with first-order elimination. Two covariates could be identified after multivariate regression analysis: VWF:Ag had a negative association with clearance and the most recently measured FVIII:C had a positive association with the estimated baseline. In table 1, the most important model characteristics are shown.

The integrated population PK model, used in this study, describes the average PK, including the variability of the baseline FVIII:C and FVIII:C response between and within patients following both the administration of desmopressin and the administration of FVIII concentrate. This model will be used for both the perioperative

**Table 1** Model characteristics of the population PK model for FVIII concentrates

| Parameter | Population estimate | RSE (%) |
|---|---|---|
| Baseline FVIII:C (IU/mL) | 0.211 | 10.9 |
| Clearance (mL/h) | 208 | 10 |
| Volume of distribution (mL) | 3400 | 4.9 |
| Proportional error (%) | 17.3 | 8.4 |
| VWF:Ag on clearance | −0.285 | 5.4 |
| Most recent FVIII:C on baseline FVIII:C | 0.891 | 15.8 |
| IOV of baseline FVIII:C (%) | 54.7 | 11.0 |
| IIV of clearance (%) | 37.4 | 19.7 |
| IIV of volume of distribution (%) | 20.8 | 26.8 |

The covariate associations between VWF:Ag and clearance was modelled as a non-linear function, while the association between most recent FVIII:C and baseline FVIII:C was modelled as a linear function. The numbers for the covariate associations (−0.285 and 0.891) describe both the shape and magnitude of the covariate effect: the more the number deviates from 0, the larger the effect of a covariate on the PK parameter.

IIV, interindividual variability; IOV, interoccasion variability; RSE, relative standard error; VWF:Ag, von Willebrand factor antigen.

setting and around bleeding episodes. The individualised dosing advice for the first dose of FVIII concentrate will be provided based on this model along with the results of the desmopressin test administration and the perioperative FVIII target levels. Further dosing of FVIII concentrates will be adjusted daily by iterative maximum a posteriori Bayesian analysis based on the population PK model in conjunction with perioperatively measured FVIII:C. In this analysis, the following information will be included: (1) FVIII release in response to desmopressin after the test dose, (2) baseline FVIII:C and (3) measured FVIII:C (both trough and peak levels) (figure 1 and table 2). Dose adjustments based on this iterative Bayesian analysis will be performed in NONMEM software. All dosing advices will be given 1 day in advance.

### Experienced quality of patient care
To assess experienced quality of care during this innovative intervention, two questionnaires will be given to the patients. Side effects will be assessed using the questionnaire previously developed and used by Stoof *et al*.[15] This questionnaire includes the occurrence of seven different self-reported side effects on a five-point scale and two on a 10-point scale. Side effects will be evaluated at two time points: before surgery and 3 days after surgery. In the second questionnaire (3 days after surgery), experienced quality of care will be evaluated with the addition of three questions dedicated to desmopressin and FVIII concentrate combination treatment. Patients will report their satisfaction with the combination treatment on a scale of 1 to 100. They will have the opportunity to explain what

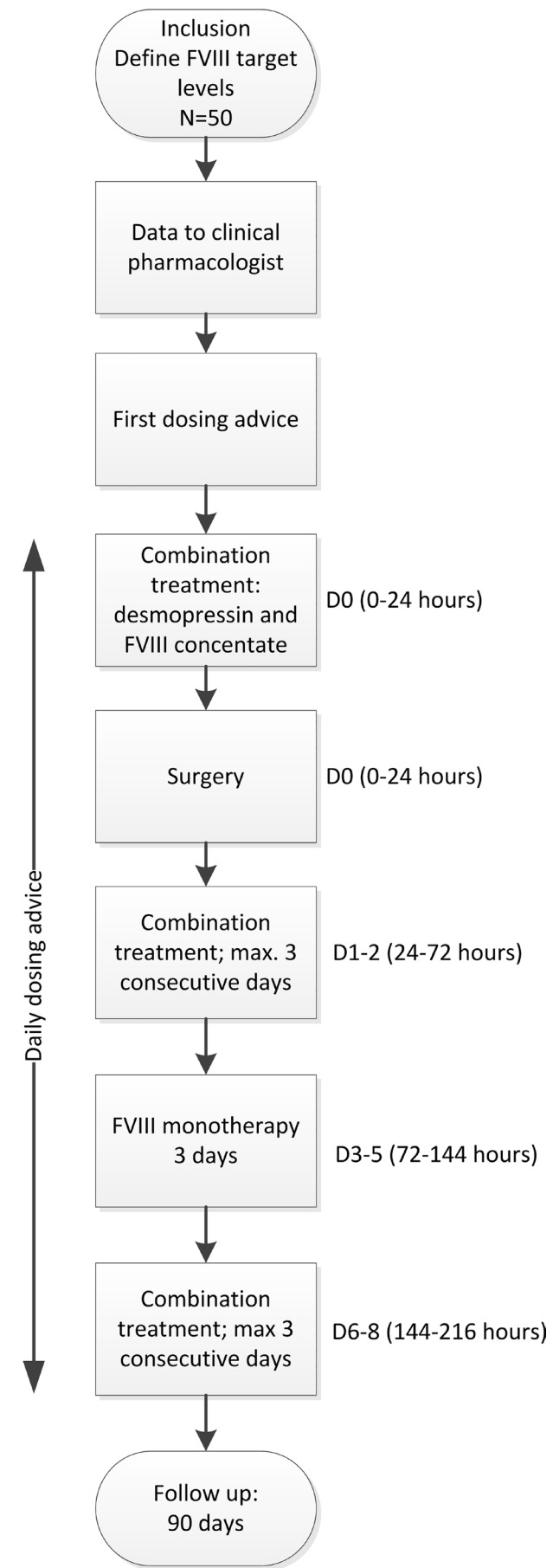

**Figure 1** Flowchart of study. FVIII, factor VIII.

**Table 2** Overview of blood sampling for FVIII:C measurements

| | Within 4weeks before surgery* | D0 = day of first desmopressin infusion Pre | D0 Post | D0 Peak | D0 Post-treat | D1 Pre | D1 Post | D1 Peak | D1 Post-treat | D2 Post | D2 Peak | D3 | 4–8 weeks after FVIII treatment |
|---|---|---|---|---|---|---|---|---|---|---|---|---|---|
| Time—bolus infusions† | Baseline | Pre | Post | Peak | Post-treat | Pre | Post | Peak | Post-treat | Post | Peak | Trough | |
| Time—continuous infusion‡ | Baseline | Pre | Post | Peak | Post-treat | Pre | Post | Peak | Post-treat | Post | Preadjust | Steady state | |
| Primary endpoint | | | | | | X | | | | X | | X | |
| Sodium | X | X | | | | X | | | | X | | X | |
| Neutralising antibodies | X | | | | | | | | | | | | X |

Grey indicates obligatory measurements.

FVIII measurements are shown here until day 3 after start of combination treatment. Monitoring will continue if treatment is still necessary.

*Only in case of elective surgery, otherwise before first desmopressin administration.

†Pre, before desmopressin; post, after desmopressin; peak, before FVIII concentrate; post-treat=after surgery; in case of bleeding: 2–6 hours after desmopressin; trough, before next dose of FVIII concentrate.

‡Pre, before desmopressin; post, after desmopressin; peak, after desmopressin; peak, after loading dose FVIII concentrate; post-treat, after surgery; in case of bleeding: 2–6 hours after desmopressin; preadjust, before dosage; steady state, FVIII measurement at random time point.

is needed to improve the given grade. Finally, they can state their preference for one of the treatment options: FVIII concentrate monotherapy or combination treatment. The last question will only be asked to patients who have a previous experience with FVIII concentrate monotherapy.

## Sample size

In the DAVID study, the proportion of patients that reach FVIII target levels with combination treatment in the first 72 hours postoperatively (without adding off-protocol FVIII concentrates) will be assessed. Historical data on current FVIII treatment show a proportion of 0.3 of patients with non-severe haemophilia A with FVIII:C within target ranges in this time period.[2] A doubling of this proportion, leading to a proportion of patients of 0.6, is believed to be clinically relevant. To study this with a power of 90% and a two-sided significance level of 0.05, a sample size of minimally 25 patients is needed. As different surgical procedures, both major and minor, may be performed on included patients and the baseline FVIII:C level may have a wide range (approximately 0.01–0.60 IU/mL), the included patient population may be heterogeneous. In addition, patients may drop out during the perioperative treatment. This may, for example, be the case if patients are no longer eligible to receive desmopressin due to changed clinical status. However, no specific stopping criteria are present. To overcome both the heterogeneity and possible dropout, we aim to include 50 patients.

To reach our target sample size and ensure we do not miss any patients, all participating centres will be updated regularly by e-mail, newsletters and during meetings.

## Outcome measures

### Primary outcome

The primary outcome will be the proportion of patients with FVIII trough levels within the FVIII target range during the first 72 hours after start of combination treatment. If a patient has one trough level outside the target range or needs off-protocol FVIII concentrate within the first 72 hours of combination the treatment, the primary endpoint is not reached in that patient. Off-protocol FVIII concentrate is defined as all administered FVIII concentrate outside the PK-guided dosing advice as given by the clinical pharmacologist. FVIII:C will be targeted according to the Dutch Treatment Guideline (table 3).[21] The treating physician may deviate from the guideline

**Table 3** Target ranges for factor VIII:C in IU/mL in the perioperative setting[21]

| Time | FVIII target level (IU/ml) |
|---|---|
| Day 0 (hour 0–24) | 0.8–1.0 (peak) |
| Day 1–4 (hour 24–96) | 0.5–0.8 (trough) |
| Day ≥5 (hour>96) | 0.3–0.5 (trough) |

and set different target ranges, based on bleeding phenotype/history or type of surgery.

FVIII:C measurements will be performed with the one-stage assay. Each participating may use its own assay. However, all centres are certified and accredited.

### Secondary outcomes
► FVIII concentrate consumption, expressed as the total amount of administered units of FVIII concentrate per kilogram per patient;
► Number and nature of adverse events during combined treatment;
► Incidence and severity of bleeding, where the severity of bleeding will be graded according to the International Society on Thrombosis and Haemostasis (ISTH) criteria for major and minor bleeding[22 23];
► Incidence of FVIII neutralising antibodies; measured with the Bethesda assay
► Incidence of thrombosis, where thrombosis will be defined according to the Dutch guidelines on thrombosis, myocardial infarctions and strokes[24 25];
► Incidence and extent of tachyphylaxis, defined as a reduction in the absolute increase in FVIII:C after the second and third desmopressin infusion;
► Medical costs and an economic evaluation;
► Experience quality of patient care, measured by a questionnaire.

### Data analysis plan
All baseline characteristics will be described as means and SD or medians with IQR, dependent on whether the parameter is normally distributed. The primary outcome, the proportion of patients within FVIII target ranges, will be analysed by a $\chi^2$ test. Retrospective data will be used as a reference, as the DAVID study only has one study arm.[2] PK data will be analysed using the NONMEM software package. Tachyphylaxis of FVIII:C response to desmopressin will be analysed with a paired t-test. Other secondary outcomes will be documented in a descriptive manner, except for the economic analysis.

### Economic analysis
An economic evaluation will be performed from a healthcare perspective from the day of surgery to 90 days postoperatively. The cost-effectiveness of combined treatment will be assessed by calculating the incremental cost-effectiveness ratio (ICER), defined as the difference in costs of combined treatment, compared with usual care, divided by the average change in effectiveness. Actual medical costs will be calculated by multiplying the volumes of healthcare use with the corresponding unit prices. Perioperative data resource use (desmopressin, FVIII concentrates, additional FVIII:C and VWF measurements, PK profiling) will be collected from medical files. Usual medical costs (to compare to combination treatment costs) will be calculated based on a treatment protocol as it would have been without desmopressin use, taking into account patient's body weight, baseline FVIII:C and type of surgery. The reduction in costs will be represented by the usual medical costs (units of FVIII concentrates) minus the actual medical costs for combined treatment (units of FVIII concentrates, µg of desmopressin, extra FVIII:C measurements). An additional sensitivity analysis will be performed to assess the stability of the results to changes in costs and effectiveness parameters. The primary effectiveness parameter is the proportion of patients reaching FVIII target levels with desmopressin and FVIII concentrate combination treatment. The secondary effectiveness parameter is the frequency of adverse events during combination treatment.

### Patient and public involvement
During the development of this study, we worked closely together with the Netherlands Haemophilia Patient Society (NVHP). One representative of the NVHP is member of the project committee and helped us with the study design. Moreover, five members of the NVHP were invited to comment on the patient information. The design and information were adjusted according to their opinions and questions. The final results of the DAVID study will be communicated through scientific international journals and at international conferences. One major conference may be that of the World Federation of Haemophilia, attended by both physicians, researchers and patients from all over the world. In addition, the results will be communicated in the magazine of the NVHP. Finally, results will also be implemented in the treatment guidelines and patient information will be adjusted accordingly.

### Proof of concept
As a proof of concept, we present one case of a patient that was treated with desmopressin and FVIII concentrate combination treatment for the duration of 24 hours. This patient has signed a BMJ consent form to publish his information in this manuscript. The patient was a 53-year-old man with moderate haemophilia A (FVIII:C 0.04 IU/ml). His body weight was 100 kg. He was in need of dental surgery and needed treatment to prevent bleeding. He received an infusion of desmopressin (0.3 µg/kg) over 30 min, followed 30 min later by FVIII concentrate in a dose of 25 IU/kg (2500 IU). He received a second dose of FVIII concentrate (20 IU/kg; i.e. 2000 IU) in the evening to maintain his FVIII:C above target values. Both FVIII concentrate dosages were determined using the integrated PK model and based on the patient's response to a previous desmopressin dose and a previous FVIII concentrate administration. Figure 2 shows the measured FVIII:C and the FVIII:C as predicted by the integrated PK model. All measured FVIII:C levels were within 0.10 IU/mL of the predicted levels (figure 2). The patient only had mild side effects of desmopressin, that is, flushing and mild tachycardia (101 bpm). To prevent bleeding, he was also treated with tranexamic acid for multiple days. No bleeding was reported before and after the surgery.

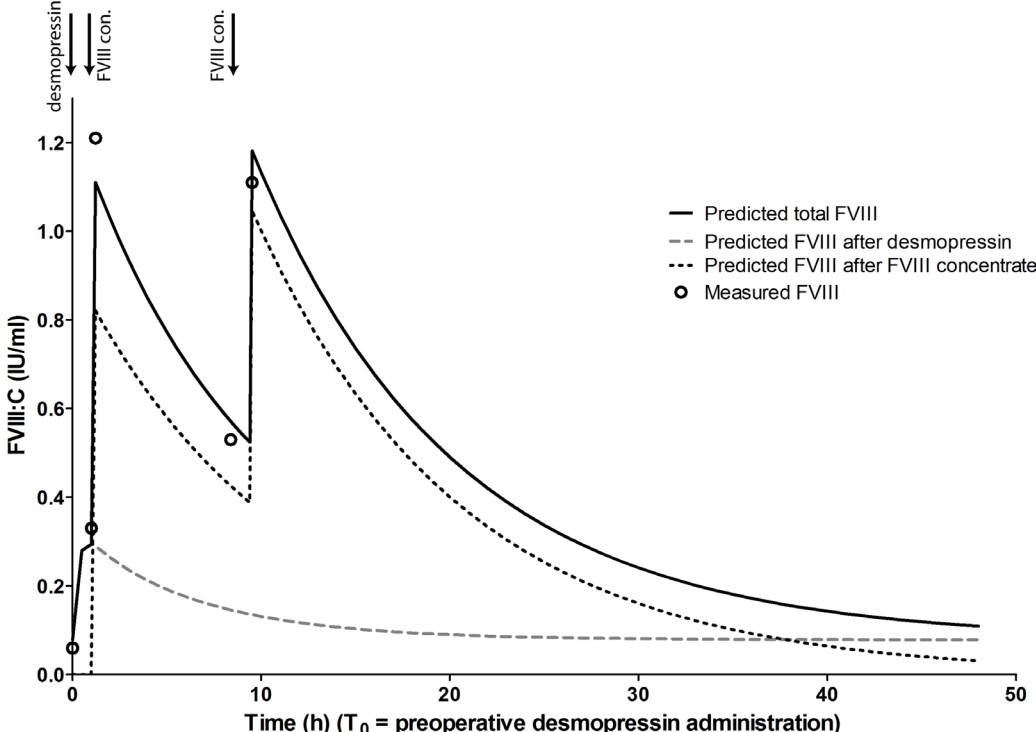

**Figure 2** Factor VIII (FVIII):C course after desmopressin and FVIII concentrate combination treatment in a pilot patient with moderate haemophilia A (FVIII:C 0.04 IU/mL). Lines are the predicted FVIII:C. Predictions were based on a previous desmopressin test dose and FVIII concentrate administration, prior to study inclusion. Solid line is total FVIII:C and can be measured. Open circles are measured FVIII:C. Infusions of desmopressin and FVIII concentrate (FVIII con.) are depicted with arrows. $T_0$, preoperative desmopressin infusion.

However, because of a possible infection of the wound, he was treated with antibiotics.

If this patient had not received combination treatment, he would have been treated with FVIII concentrate monotherapy both before and after the dental surgery. As his baseline FVIII:C was 0.04 IU/mL, his body weight 100 kg and the target FVIII peak level was 0.80–1.00 IU/mL, the FVIII loading dose would be (1.00–0.04)/0.02*100=4750 IU (rounded to entire vials). In the evening, the patient would have received 2500 IU to maintain the FVIII:C between the target FVIII. Therefore, combination treatment hypothetically saved 2750 IU FVIII concentrate in this patient.

### ETHICS AND DISSEMINATION

The study was approved. The study will be conducted according to good clinical practice (GCP) guidelines and the Declaration of Helsinki. Also, see online supplementary data 3–6 for our regulations for data storage, amendments and compensation for injury. Written informed consent will be obtained from all patients by a member of the research team (see online supplementary data 7 for the patient information, Dutch only). Results of the study will be communicated to the (inter)national medical and scientific community trough publication in high-ranking peer-reviewed international journals and at (inter) national medical scientific conferences. Results of the study will also be implemented in the Dutch Haemophilia

Treatment Guidelines. Hopefully, the international society of Haemophilia Treatment Centres will adapt the results in their guidelines as well.

### Data monitoring committee and serious adverse events

This study does not carry any large safety risks as both FVIII concentrates and desmopressin are registered therapeutics for haemophilia treatment. In addition, to guarantee safety for all patients in this study, FVIII:C levels will be closely monitored to prevent any additional bleeding risks. Therefore, a data safety monitoring board is not needed.

Serious adverse events (SAE) will be communicated to the sponsor within 24 hours. The sponsor will register the SAE within 15 days on ToetsingOnline, the Dutch registration system for SAEs.

### REGISTRATION

The trial is registered in the Dutch Trial Registry, number NTR5383 (www.trialregister.nl) and in EudraCT: 2014-00535-14.

### DISCUSSION

This prospective trial will allow us to evaluate the safety and efficacy of combination treatment of desmopressin and FVIII concentrate in reaching target FVIII:C during bleeding episodes and in the perioperative setting.

Dosing of FVIII concentrate will be determined by an integrated population PK model developed specifically for patients with non-severe haemophilia A to be treated with combination treatment. Fifty patients with non-severe haemophilia A will be included from eight haemophilia treatment centres in the Netherlands to reach our aim.

The DAVID study has some limitations. First, this study is not designed as a randomised controlled trial; it does not include a control group. Therefore, no direct comparison to standard treatment will be possible. However, we performed an extensive retrospective cohort study to determine the effectiveness of current clinical practice in which 37 patients undergoing 52 surgeries were evaluated.[2] Moreover, the amount of FVIII concentrate that would have been administered to the patients included in the DAVID study, as if desmopressin was not administered and as if no PK-guided dosing was applied, will be calculated.

The second limitation is the heterogeneity of the study population. All types of surgery, unless not compatible with desmopressin treatment, may be included in the study. To limit the effects of this heterogeneity, only patients with an expected treatment duration of at least 48 hours will be included. Moreover, the data that were used to develop the integrated population PK model also included data from various types of surgery.

## CONCLUSION

In the DAVID study, efficacy and safety of desmopressin and FVIII concentrate combination treatment in patients with non-severe haemophilia A will be determined. Using this innovative approach treatment of patients with non-severe haemophilia A both during bleeding episodes and in the perioperative setting may be improved.

#### Author affiliations
[1]Department of Haematology, Erasmus University Medical Centre, Rotterdam, The Netherlands
[2]Department of Paediatric Haematology, Erasmus University Medical Centre-Sophia Children's Hospital, Rotterdam, The Netherlands
[3]Department of Hospital Pharmacy–Clinical Pharmacology Unit, Amsterdam UMC, University of Amsterdam, Amsterdam, The Netherlands
[4]Netherlands Haemophilia Patient Society, (NVHP), Nijkerk, The Netherlands
[5]Department of Paediatric Haematology, Amsterdam UMC, Emma Children's Hospital, Amsterdam, The Netherlands
[6]Department of Plasma Proteins, Sanquin Research, Amsterdam, The Netherlands
[7]Department of Public Health, Erasmus University Medical Centre, Rotterdam, The Netherlands
[8]Department of Haematology, Maastricht University Medical Centre, Maastricht, The Netherlands
[9]Department of Vascular Medicine, Amsterdam UMC, University of Amsterdam, Amsterdam, The Netherlands
[10]Department of Thrombosis and Haemostasis, Leids Universitair Medisch Centrum, Leiden, The Netherlands
[11]Department of Haematology, Radboud University Medical Centre, Nijmegen, The Netherlands
[12]Department of Haematology, Universitair Medisch Centrum Groningen, Groningen, The Netherlands
[13]Department of Haematology, Maxima Medical Centre, Eindhoven, Eindhoven, Noord-Brabant, The Netherlands
[14]Van Creveldclinic, University Medical Centre Utrecht, Utrecht, The Netherlands

**Correction notice** This article has been corrected since it first published online. The open access licence type has been amended.

**Acknowledgements** We would like to specially thank MHED for her contribution on behalf of the NVHP and the five NVHP members who helped to improve the patient information. In addition we would like to thank the pilot patient for his cooperation and for letting us use his case as an example.

**Contributors** MJHAK, MHC, FWGL and RAAM designed the study and critically revised the manuscript. LMS, SP and RMvH wrote the manuscript and refined the study design. MHED, KF, EAMB, MC, JE, BAPL-vG, KM, LN and EPM-B critically revised the manuscript and refined the study design.

**Funding** This work was supported by ZonMw, (grant number 836031003), the Dutch organisation for Health Research and Development.

**Competing interests** LMS: received reimbursement from CSL-Behring for attending a symposium, not related to this study. MHC: received unrestricted research/educational funding for various projects as well as travel fees from the following institutions and companies: ZonMW, Innovatiefonds, Pfizer, Baxalta/Shire, Bayer Schering Pharma, Novo Nordisk, Novartis, Roche and CSL Behring, all not related to this study. RMvH, EAMB, MC, MHED, LN, SP: nothing to disclose relevant to the DAVID study. KF: is a member of the European Hemophilia Treatment and Standardization Board sponsored by Baxter, has received unrestricted research grants from CSL Behring and Bayer and has given lectures at educational symposiums organized by Pfizer, Bayer and Baxter. JE: received research funding from CSL Behring and honorarium for educational activity from Roche, not related to this study. BAPL-vG: received unrestricted educational grants from Baxter and CSL Behring and speaker fees from Sanquin. KM: research support from Bayer, Sanquin and Pfizer; speaker fees from Bayer, Sanquin, Boehringer Ingelheim, BMS and Aspen; consulting fees from Uniqure, not related to this study; FWGL: received unrestricted research grants from CSL-Behring and Baxalta/Shire not related to this study. He is consulant for Shire, NovoNordisk and UniQure. Fees go to the university. RAAM: received personal fees from Merck Sharp & Dohme and Zeria and grants from Bayer, UCB Pharma and Hoffman La Roche with no involvement in this study. MJHAK: received unrestricted research grants from Pfizer, Innovatiefonds and Ferring with no involvement in this study.

**Patient consent for publication** Obtained.

**Ethics approval** Medical Ethics Committee of the Erasmus University Medical Centre Rotterdam, the Netherlands.

**Provenance and peer review** Not commissioned; externally peer reviewed.

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
