## [Reviewer comments · BMJ Open]

ARTICLE DETAILS

TITLE (PROVISIONAL)	Desmopressin treatment combined with clotting factor VIII concentrates in patients with non-severe haemophilia A: protocol for a multicentre single-armed trial, the DAVID study
AUTHORS	Schütte, Lisette; Cnossen, Marjon; Van Hest, Reinier; Driessens, Mariette; Fijnvandraat, Karin; Polinder, Suzanne; Beckers, Erik; Coppens, Michiel; Eikenboom, Jeroen; Laros-van Gorkom, Britta; Meijer, Karina; Nieuwenhuizen, Laurens; Mauser-Bunschoten, Evelien; Leebeek, Frank; Mathôt, Ron; Kruij, Marieke

VERSION 1 – REVIEW

REVIEWER	Andrea Edginton University of Waterloo, Canada
REVIEW RETURNED	14-Apr-2018

GENERAL COMMENTS	This protocol outlines a proposal to assess the efficacy and safety of combined desmopressin + FVIII administration to moderate hemophilia A patients undergoing surgery or during a bleed event. Major comment: There is no reference for the integrated popPK model or description of the model structure in this protocol. If there is a reference, even if in abstract form, please include. I am particularly interested in the error associated with Bayesian forecasting following desmopressin administration when taking only a predose, 1 and 4 hour FVIII level and the correlation between baseline and FVIII release following desmopressin administration. I recognize that this is a protocol paper however study design here includes the model for which there are no details. Minor comments: The group has previously published a popPK model for the perioperative scenario. They have not done this for bleeding from (minor) trauma. Will the researchers be using the same model regardless of scenario? No details on the dose scaling algorithm are provided. Please explain. “Finally, they can state their preference for one of the treatment options: FVIII concentrate monotherapy or combination treatment” This is only valid if the patient has a benchmark from which to compare (e.g. has had previous monotherapy). Administrative comments: Inclusion criteria state that results of a desmopressin test must be available and provide a minimal increase of >0.2 IU/mL. Later, it is
---

	explained that in the event no test is available, the researchers will perform one. It however states “When no desmopressin test has been performed meeting our criteria, the test should be performed...” Does this suggest that if a previous test doesn’t meet the >0.2 IU/mL cutoff, another test will be performed? The researcher suggest that the test is stable over time based on age-since-test criteria. Consider removing “...meeting our criteria...”. And replacing with “meeting this criteria..” if the criteria to which is referred is the timing criteria. The popPK model that will be used is an integrated model with respect to “endogenous FVIII:C after desmopressin administration and exogenous FVIII:C after FVIII concentrate administration”. I suggest stating that the exogenous model is really exogenous plus endogenous as baseline must be a component of the model. “The patient did not suffer from any other than mild side effects of desmopressin...” There is a word(s) missing. “We aim to prove that the proportion of patients with FVIII:C within the FVIII target ranges can be increased by the use of the combination...” Please reword as hypotheses cannot be proven.
--	--

REVIEWER	Maissaa Janbain Tulane School of Medicine, Louisiana Center for Bleeding and Clotting Disorders New Orleans, LA, USA
REVIEW RETURNED	28-May-2018

GENERAL COMMENTS	A more thorough explanation of sample size calculation is needed, drop out, and heterogeneity assumptions... An inclusion of the economic analysis in proof of concept paragraph is helpful. Should they also include requirement to use other hemostatic agents like antifibrinolytics.
---

VERSION 1 – AUTHOR RESPONSE

Reviewers' Comments to Author:

Reviewer: 1

Reviewer Name: Andrea Edginton

Institution and Country: University of Waterloo, Canada Competing Interests: none declared

This protocol outlines a proposal to assess the efficacy and safety of combined desmopressin + FVIII administration to moderate hemophilia A patients undergoing surgery or during a bleed event.

Major comment:

There is no reference for the integrated popPK model or description of the model structure in this protocol. If there is a reference, even if in abstract form, please include. I am particularly interested in the error associated with Bayesian forecasting following desmopressin administration when taking only a predose, 1 and 4 hour FVIII level and the correlation between baseline and FVIII release following desmopressin administration. I recognize that this is a protocol paper however study design here includes the model for which there are no details.

In answer to your comment, the model used to calculate dosages of FVIII concentrate is an integration of two population PK models: 1) a population PK model of FVIII response after

desmopressin administration and 2) a population PK model of FVIII after administration of FVIII concentrate. We added the reference for the population model of FVIII after desmopressin administration to the manuscript. (page 8, line 16) As the model for FVIII after administration of FVIII concentrate has not been published yet, we provided the main characteristics of the model in the manuscript (page 8) and in table 1.

Minor comments:

The group has previously published a popPK model for the perioperative scenario. They have not done this for bleeding from (minor) trauma. Will the researchers be using the same model regardless of scenario?

The same model will be used for both scenarios. This has been clarified in the manuscript:

This model will be used for both the perioperative setting and around bleeding episodes. (page 8, line 28-29)

The reason we did this, is that no separate model is available for bleedings. However, in both the population PK model of FVIII response after desmopressin administration and the population PK model of FVIII after administration of FVIII concentrate, patients with bleeding episodes were included. Therefore we believe, the use of these models is justified.

No details on the dose scaling algorithm are provided. Please explain.

There is not one dose scaling algorithm. FVIII concentrate dosages will be calculated based on the population PK model in conjunction with perioperatively measured FVIII levels, using maximum a posteriori Bayesian estimation, as implemented in the NONMEM software package. This is explained in the manuscript (page 8+9)

“Finally, they can state their preference for one of the treatment options: FVIII concentrate monotherapy or combination treatment” This is only valid if the patient has a benchmark from which to compare (e.g. has had previous monotherapy).

Thank you for your comment. We added to the manuscript that this question will only be asked to patients who have had a previous experience with FVIII concentrate monotherapy:

The last question will be only asked to patients who have a previous experience with FVIII concentrate monotherapy. (page 9, line 17+18)

Administrative comments:

Inclusion criteria state that results of a desmopressin test must be available and provide a minimal increase of >0.2 IU/mL. Later, it is explained that in the event no test is available, the researchers will perform one. It however states “When no desmopressin test has been performed meeting our criteria, the test should be performed...” Does this suggest that if a previous test doesn’t meet the >0.2 IU/mL cutoff, another test will be performed? The researcher suggest that the test is stable over time based on age-sincetest criteria. Consider removing “...meeting our criteria...”. And replacing with “meeting this criteria..” if the criteria to which is referred is the timing criteria.

We agree the current text is not clear regarding this matter. If a patient has had an insufficient previous response, the test will not be repeated purely for our study and the patient will be excluded. Therefore the sentence was changed to:

“meeting these criteria..” (page 8, line 5). Hereby we refer only to the timing and number of FVIII:C measurements.

The popPK model that will be used is an integrated model with respect to “endogenous FVIII:C after desmopressin administration and exogenous FVIII:C after FVIII concentrate administration”. I suggest stating that the exogenous model is really exogenous plus endogenous as baseline must be a component of the model.

We changed the text according to your comment to: *“The integrated population model, used in this study, describes the average PK, including the variability of the baseline FVIII:C and FVIII:C response*

between and within patients following both the administration of desmopressin and the administration of FVIII concentrate.”(page 8, line 26-28)

“The patient did not suffer from any other than mild side effects of desmopressin...” There is a word(s) missing.

As this sentence was not clear, it was changed to: *“The patient only had mild side effects of desmopressin, i.e. flushing and mild tachycardia (101 bpm).”* (page 12, line 17-18)

“We aim to prove that the proportion of patients with FVIII:C within the FVIII target ranges can be increased by the use of the combination...” Please reword as hypotheses cannot be proven.

The word *prove* was deleted and changed to *show*. (page 5, line 7-8)

Reviewer: 2

Reviewer Name: Maissaa Janbain

Institution and Country: Tulane School of Medicine, Louisiana Center for Bleeding and Clotting Disorders New Orleans, LA, USA Competing Interests: none

A more thorough explanation of sample size calculation is needed, drop out, and heterogeneity assumptions...

We extended the sample size calculation with regard to the heterogeneity and drop outs (page 9, line 27-33)

An inclusion of the economic analysis in proof of concept paragraph is helpful.

We added a calculation of the hypothetical extra FVIII concentrate, that would have been needed, if no desmopressin was used:

*If this patient would not have received combination treatment, he would have been treated with FVIII concentrate monotherapy both before and after the dental surgery. As his baseline FVIII:C was 0.04 IU/mL, his body weight 100 kg and the target FVIII peak level was 0.80-1.00 IU/mL, the FVIII loading dose would be $(1.00-0.04)/0.02*100=4750$ IU (rounded to entire vials). In the evening the patient would have received 2500 IU to maintain the FVIII:C between the target FVIII. Therefore, combination treatment hypothetically saved 2750 IU FVIII concentrate in this patient. (page 12)*

Should they also include requirement to use other hemostatic agents like antifibrinolytics.

Patients included in the DAVID-study may receive all needed comedication, including other haemostatic agents. We agree it was not stated clearly whether comedication could be used, while this is important in haemophilia treatment. Therefore we added a subheading: *concomitant treatment* (page 7):

As patients receive desmopressin, they will have a fluid restriction of 1.5 L per 24 hours, till 24 hours after the last desmopressin administration. Furthermore, all concomitant treatment and therapeutics are allowed, except for the therapeutics specified in the exclusion criteria. This includes other treatment than desmopressin or factor concentrate necessary to prevent or treat bleeding, such as tranexamic acid. The same will apply for prophylactic treatment to prevent thrombosis. These treatment options will be applied at the discretion of the treating physician and will be documented in all patients.

In case a patient uses comedication that has an interaction with desmopressin, the patient will not be included in the study as we stated that in our exclusion criteria.

VERSION 2 – REVIEW

REVIEWER	Andrea Edginton School of Pharmacy, University of Waterloo, Canada
REVIEW RETURNED	17-Jul-2018

GENERAL COMMENTS	"The within-patient variability (variability between different treatment episodes with FVIII concentrate) was 51.2% on baseline FVIII:C. " Note that the value of 51.2% is not in Table 1 and Table 1 is IIV and not IOV as suggested by the above sentence. Baseline should not be modeled with IOV as it does not change with occasion; it is an inherent patient steady state value. I don't think this was done but I think there are inconsistencies in the text and table.
--

VERSION 2 – AUTHOR RESPONSE

Reviewer's Comments to Author:

Reviewer: 1

Reviewer Name: Andrea Edginton

Institution and Country: School of Pharmacy, University of Waterloo, Canada Please state any competing interests or state 'None declared': None declared

"The within-patient variability (variability between different treatment episodes with FVIII concentrate) was 51.2% on baseline FVIII:C. " Note that the value of 51.2% is not in Table 1 and Table 1 is IIV and not IOV as suggested by the above sentence. Baseline should not be modeled with IOV as it does not change with occasion; it is an inherent patient steady state value. I don't think this was done but I think there are inconsistencies in the text and table.

Thank you for noticing the difference between the text and the table. Indeed, in the text as well as in the table, the results of the IOV should be given. To prevent any confusion, we adjusted the table and deleted above sentence from the text as the results can already be found in Table 1. Although the baseline of a parameter does not always have IOV, FVIII:C can change over time in each patient and therefore has IOV. Moreover, in this model we found the IOV of baseline FVIII:C was even more important than the IIV of FVIII:C. In fact, by incorporating the IOV in the model no IIV of FVIII:C was left anymore.